Thrombocytopenia and increased risk of adverse outcome in COVID-19 patients

Yuan Yang 1
Wang Gang 1
Chen Xi 2
Ye Xiao-Lei 1
Li Xiao-Kun 1
Li Rui 3 4
Jiang Wan-Li 5
Zeng Hao-Long 6
Du Juan 7
Zhang Xiao-Ai 1
Li Hao 1
Fang Li-Qun 1
Lu Qing-Bin qingbinlu@bjmu.edu.cn qingbinlu@vip.qq.com 7
Liu Wei lwbime@163.com liuwei@bmi.ac.cn 1 7
1 State Key Laboratory of Pathogen and Biosecurity, Beijing Institute of Microbiology and Epidemiology , Beijing , China
2 Department of Thoracic and Vascular Surgery, Wuhan First Hospital, Tongji Medical College, Huazhong University of Science and Technology , Wuhan , China
3 Department of Healthcare, School of Health Sciences, Wuhan University, Wuhan, Hubei , Wuhan , China
4 Global Health Institute, Wuhan University, Wuhan, Hubei , Wuhan , China
5 Department of Thoracic Surgery, Renmin Hospital of Wuhan University , Wuhan , China
6 Department of Laboratory Medicine, Tongji Hospital, Tongji Medical College, Huazhong University of Science and Technology , Wuhan , China
7 Department of Laboratorial Science and Technology, School of Public Health, Peking University , Beijing , China
Fujioka Kazumichi
Electronic publication date: 2022 Jun 30
Publication date: 2022
Volume: 10
Electronic Location ID: e13608
Received 2021 Aug 24; Accepted 2022 May 27
Copyright: ©2022 Yuan et al.
Copyright year: 2022
Copyright holder: Yuan et al.
License: This is an open access article distributed under the terms of the Creative Commons Attribution License, which permits unrestricted use, distribution, reproduction and adaptation in any medium and for any purpose provided that it is properly attributed. For attribution, the original author(s), title, publication source (PeerJ) and either DOI or URL of the article must be cited.
License URL: https://creativecommons.org/licenses/by/4.0/

Keywords: Thrombocytopenia, COVID-19, Adverse outcome, Mortality, China

Funding: China Mega-Project for Infectious Diseases grant 2018ZX10713002 2018ZX10101003 2017ZX10103004 Natural Science Foundation of China 81825019 This study was supported by the China Mega-Project for Infectious Diseases grant (2018ZX10713002, 2018ZX10101003, 2017ZX10103004) and the Natural Science Foundation of China (81825019). The funders had no role in study design, data collection and analysis, decision to publish, or preparation of the manuscript.

==============================
Background

Thrombocytopenia was common in the coronavirus disease 2019 (COVID-19) patients during the infection, while the role of thrombocytopenia in COVID-19 pathogenesis and its relationship with systemic host response remained obscure. The study aimed to systematically evaluate the relationship between thrombocytopenia in COVID-19 patients and clinical, haematological and biochemical markers of the disease as well as adverse outcomes.

Methods

To assess the relationship between abnormal platelet levels and disease progression, a multi-center retrospective cohort study was conducted. COVID-19 patients with thrombocytopenia and a sub-cohort of matched patients without thrombocytopenia were compared for their clinical manifestations, haematological disorders, biochemical parameters, inflammatory markers and clinical outcome.

Results

Thrombocytopenia was present in 127 of 2,209 analyzed patients on admission. Compared with the control group, thrombocytopenia patients developed significantly higher frequency of respiratory failure (41.9% vs. 22.6%, P = 0.020), intensive care unit entrance (25.6% vs. 11.5%, P = 0.012), disseminated intravascular coagulation (45.2% vs. 10.6%, P < 0.001), more altered platelet morphology indexes and coagulation perturbation, higher levels of inflammatory markers. In addition, a significantly increased all-cause mortality (hazard ratio 3.08, 95% confidence interval 2.26–4.18, P < 0.001) was also observed in the patients with thrombocytopenia. Late development of thrombocytopenia beyond 14 days post-symptom was observed in 61 patients, from whom a comparable mortality rate yet longer duration to death was observed compared to those with early thrombocytopenia.

Conclusions

Our finding from this study adds to previous evidence that thrombocytopenia is associated with adverse outcome of the disease and recommend that platelet count and indices be included alongside other haematological, biochemical and inflammatory markers in COVID-19 patients’ assessment during the hospital stay.

Introduction

Coronavirus disease 2019 (COVID-19) caused by a novel RNA virus (SARS-CoV-2) has widely spread to become the greatest public health challenge to date. Up to 5 August 2021, it had affected over 200,000,000 individuals in more than 200 countries and territories worldwide causing over 4,300,000 deaths, severely stressed health systems and the global economy (World Health Organization, 2021). In the majority of cases, COVID-19 is a mild illness, while some develop severe disease and a minority of patients develop critical disease with septic shock or respiratory and/or multi-organ failure (Wu & McGoogan, 2020). Laboratory tests have been assessed with regard to their value as prognostic indicators. Among all the laboratory indicators, the hematologic parameters have been intensively described, especially with lymphopenia commonly established with significance for their prognostic capacity for severe or deadly outcome after SARS-COV-2 infection (Frater et al., 2020). Both CD4 and CD8 counts decreased during the early course and were associated with adverse outcomes. Leukocytosis appears to herald bacterial infection or superinfection (Lippi & Plebani, 2020) and neutrophilia is an expression of the cytokine storm and hyperinflammatory state (Chen et al., 2020; Huang et al., 2020; Mehta et al., 2020; Qin et al., 2020), both acting as important pathogenetic markers in COVID-19. Thrombocytopenia, a common feature in viral hemorrhagic fever virus, has also been reported in respiratory virus infection and several well-studied viruses, such as human immunodeficiency virus, influenza virus, cytomegalovirus, Ebola virus and rhinovirus (Assinger, 2014). Thrombocytopenia has been described in two other known human coronaviruses, with a proportion of 55% in the SARS (Frater et al., 2020), and 36% in the middle east respiratory syndrome (Assiri et al., 2013), and recorded in the episode of emerging SARS-COV-2 infection. Although previous evidence indicating that platelets (PLT) can influence disease outcome is abundant, clinical studies addressing the key host responses in COVID-19 related thrombocytopenia are limited. The detailed information on the relationship between PLT and other clinical outcomes/presentations was absent. Here we investigated a large cohort of COVID-19 patients who were consecutively observed, with the aim to systematically evaluate the relationship between thrombocytopenia in COVID-19 patients and clinical, haematological and biochemical markers of the disease as well as adverse outcomes.

Materials & Methods

Study design and patient enrollment

A retrospective cohort study was conducted in four designated hospitals for COVID-19 in Hubei province in China, including Wuhan First Hospital, Renmin Hospital of Wuhan University, Tongji Hospital, Tongji Medical College in Wuhan city, and Huangmei hospital in Huanggang city. All the patients were confirmed by positive SARS-COV-2 RNA detection on nasopharyngeal swab samples that were collected for real-time reverse transcriptase polymerase chain reaction (RT-PCR) assay (Chinese Center for Disease Control and Prevention, 2020).

Sample collection and data extraction

All of the electronic medical records of the hospitals were reviewed by trained medical staff, from which demographic information, epidemiological history, underlying comorbidities, clinical manifestations, laboratory test, CT scan, treatment history and outcome were extracted and summarized by standardized EpiData data collection form. The outcome included respiratory failure, intensive care unit (ICU) entrance, acute heart injury, acute kidney injury, septic shock and disseminated intravascular coagulation (DIC). The plasma with 0.109 mol/L sodium citrate was collected for the detection of coagulation parameters. The serum was collected for the detection of cytokines. The study was approved by the Research Ethics Commission of Tongji Hospital, Tongji Medical College (TJ-IRB20200102), and the requirement for informed consent was waived by the Ethics Commission.

Definition

Thrombocytopenia was defined as PLT < 100 × 109/L and non-thrombocytopenia was defined as the PLT ≥ 100 × 109/L which were evaluated on admission. Respiratory failure was defined as arterial oxygen partial pressure <60 mmHg with or without carbon dioxide partial pressure >50 mmhg under the condition of breathing air at rest. According to the KDIGO Clinical Practice Guideline, acute kidney injury was defined as any of the following: an increase in serum creatinine levels by ≥0.3 mg/dl (≥26.5 µmol/l) within 48 h; increase in serum creatinine to ≥1.5 times baseline, which was known or presumed to have occurred within the prior 7 days; urine volume <0.5 ml/kg/h for 6 h. Acute heart failure was defined as a clinical syndrome of current or previous symptoms and/or signs caused by structural and/or functional abnormalities of the heart, combined with elevated levels of natriuretic peptides or objective evidence of cardiogenic pulmonary or systemic congestion (Bauersachs et al., 2022). Septic shock was defined as persistent hypotension requiring vasopressors to maintain MAP ≥ 65 mm Hg and serum lactate levels > two mmol/L (18 mg/dL) on the basis of adequate volume resuscitation (Singer et al., 2016). The International Society on Thrombosis and Haemostasis (ISTH) DIC 8-point score was applied for DIC calculation (Taylor Jr et al., 2001). A platelet count >100 × 109/L accounts for 0 points, a score between 50–100 × 109/L for 1 point and a platelet count <50 × 109/L for 2 points. Prolonged prothrombin time, results in 0 points when <3 s, 1 point when 3–6 s and 2 points when ≥6 s. Fibrinogen levels ≥1 g/L result in 0 points, <1 g/L in 1 point. D-dimer levels of <0.4 µg/mL account for 0 points, levels between 0.4 and 4 µg/mL account for 2 points and D-dimer levels >4 µg/mL account for 3 points. A score ≥5 is compatible with overt DIC.

Statistical analysis

Propensity score matching (PSM) was performed (1:6 for thrombocytopenia versus non-thrombocytopenia patients), with age, sex and days from symptom onset to admission used for propensity score by the nearest-neighbor method with a caliper of 0.2 standard deviation. The standardized mean difference (SMD) was measured to determine the balance between the two groups before and after PSM, with its cut-off value of <0.2 regarded as indicating an adequate balance between two groups. The inter-group comparison of patients’ characteristics was made for the matched patients to ensure the two groups were well balanced.

Descriptive statistics were performed for categorical variables as frequencies and proportions or rates, as well as for continuous variables were expressed as medians and interquartile ranges. A Mann–Whitney U test or a Kruskal-Wallis test was used to compare the differences of continuous nonparametric variables between different groups. Pearson chi-square test or Fisher exact test was performed to compare the differences of categorical variables. Kaplan–Meier method was used to assess the survival rates, and the differences between groups with different platelet counts were analyzed by the log-rank test. Hazard ratio (HR) with 95% confidence interval (CI) were estimated using the Cox regression model, adjusting for baseline variables. Cox regression model was used to perform multivariable analysis between patients with early and late development of thrombocytopenia at 1–28 days and 28–56 days, respectively. A generalized estimating equation (GEE) was used to perform the comparisons on the levels of the parameters among two groups due to the repeated measurements in the same patient. Odds ratio (OR) with 95% CI were estimated by multivariable logistic regression model, to investigate the association between groups with different platelet counts and fatal outcome. Statistical analysis was performed using R software (version 3.6.3; R Core Team, 2020), and a two-sided P value of less than 0.05 was considered statistically significant.

Results

Patient recruiting and demographic information

From 3 January to 10 March 2020, 2346 hospitalized COVID-19 patients were enrolled, among whom 137 patients were excluded because of incomplete or missing data on the platelet count or other important parameters during hospitalization. Among the remaining 2209 COVID-19 patients available, 127 (5.7%) were recorded with thrombocytopenia, including 34 females and 93 males, with median age of 67 (IQR 56–75) years old. Compared with 2082 (94.3%) patients with PLT ≥ 100 × 109/L on admission, the thrombocytopenia patients were older, more likely to be male gender, and with higher presence of hypertension, clinical manifestation of fever, fatigue and anorexia (Table S1).

For the thrombocytopenia group, the median PLT counts were kept below the normal range throughout the observation, with the nadir level observed at three weeks after disease and increased thereafter until attaining near 100 × 109/L at four weeks after disease. For the non-thrombocytopenia patients, the median PLT counts were kept within normal range during the whole observation, and with the peaking counts observed at 15 and 18 days after disease, respectively (Fig. 1A).

Figure 1 The dynamic profile of platelet counts and 60-day Kaplan–Meier survival curves of COVID-19 patients stratified by platelet counts on admission.

(A) Dynamic profile of platelet count was plotted in the thrombocytopenia group and the non-thrombocytopenia group classified by the platelet count on admission. The median and interquartile range were shown. (B) Survival curves on probability of survival from the thrombocytopenia group and the non-thrombocytopenia group and compared by log-rank test. The analysis was performed based on the PSM database.

The PLT counts were higher in survival patients than in fatal patients by the GEE model, after adjusting the variables of age, sex, delay from symptom onset to admission and comorbidities (P < 0.001, Fig. S1A). However, the patients with comorbidities had comparable levels of PLT counts at the different stages to those without (Figs. S1B–S1E).

The comparison of clinical features based on the PSM cohort

We used PSM based on age, sex and days from symptom onset to admission to create a matched group with PLT within normal range on admission (729 patients) in Table 1. The age, sex, and days from symptom onset to admission were comparable after matching (all SMD < 0.2). Pre-existing hypertension, diabetes and coronary heart disease were non-significantly different between the two groups. The major symptoms were fever, cough, fatigue, anhelation, anorexia, and chest tightness, among which fever, fatigue and anorexia were reported with significantly higher frequencies in the thrombocytopenia group than the non-thrombocytopenia group (P = 0.019, P = 0.024 and P = 0.007, respectively).

Table 1 Basic information of patients throughout the course of the COVID-19 on the PSM database.

Characteristics	Total	Thrombocytopenia	Non-thrombocytopenia	P value	
	(N = 856)	(n = 127)	(n = 729)		
Age, years (median, IQR)	66 (56–75)	67 (56–75)	66 (56–75)	0.459*	
≤45	104 (12.1)	13 (10.2)	91 (12.5)	0.566**	
45–60	170 (19.9)	29 (22.8)	141 (19.3)		
>60	582 (68.0)	85 (66.9)	497 (68.2)		
Sex, male, n (%)	610 (71.3)	93 (73.2)	517 (70.9)	0.596**	
Days from disease onset to clinic visit, median (IQR)	
	9 (5–13)	8 (5–12)	9 (5–13)	0.441*	
Length of stay, days, median (IQR)	
	18 (11–25)	14 (7–21)	18 (12–25)	<0.001*	
Any comorbidity, n (%)	421 (49.2)	68 (53.5)	353 (48.4)	0.287**	
Hypertension	332 (38.8)	55 (43.3)	277 (38)	0.257**	
Diabetes	130 (15.2)	23 (18.1)	107 (14.7)	0.320 **	
Coronary heart disease	87 (10.2)	14 (11.0)	73 (10.0)	0.728**	
Cerebral infarction	61 (7.1)	8 (6.3)	53 (7.3)	0.695**	
Symptoms, n (%)					
Fever	651 (76.1)	107 (84.3)	544 (74.6)	0.019**	
Cough	543 (63.4)	80 (63.0)	463 (63.5)	0.911**	
Fatigue	289 (33.8)	54 (42.5)	235 (32.2)	0.024**	
Anhelation	228 (26.6)	37 (29.1)	191 (26.2)	0.490**	
Anorexia	221 (25.8)	45 (35.4)	176 (24.1)	0.007**	
Chest tightness	198 (23.1)	33 (26.0)	165 (22.6)	0.409**	
Diarrhea	83 (9.7)	5 (3.9)	78 (10.7)	0.017**	
Nausea	66 (7.7)	10 (7.9)	56 (7.7)	0.940**	
Headache	14 (1.6)	3 (2.4)	11 (1.5)	0.749**	
Notes.

IQR interquartile range

PSM propensity score matching

COVID-19 coronavirus disease 2019

* P value was calculated by Mann–Whitney U test.

** P value was calculated by Pearson chi-square test.

Comparison of all-cause mortality and detrimental complications based on the PSM cohort

Patients with thrombocytopenia developed a higher prevalence of respiratory failure (41.9% vs. 22.6%), ICU entrance (25.6% vs. 11.5%), acute heart injury (9.3% vs. 5.8%), acute kidney injury (4.7% vs. 2.9%), septic shock (9.3% vs. 2.9%), and DIC (45.2% vs. 10.6%) compared to the matched non-thrombocytopenia group. The difference remained significant for respiratory failure, ICU entrance and DIC after adjusting the effect from age, sex, days from symptom onset to admission and comorbidities (Table 2). However, there was no significant difference in the risk of acute heart injury, acute kidney injury, and septic shock between thrombocytopenia group and non-thrombocytopenia group. These results were in line with those obtained from the database without PSM (Table S2).

Table 2 Multivariable analysis on the associations between thrombocytopenia and adverse outcome for COVID-19 patients using the PSM data.

The association between thrombocytopenia and adverse outcome for COVID-19 were calculated by the multivariable logistic regression models adjusted for age, sex, delay from symptom onset to admission and comorbidities. The number of patients included in multivariable analysis was inconsistent with that of the PSM database because the clinical outcome was not obtained from all the included patients.

Parameter	Thrombocytopenia	Non-thrombocytopenia	Multivariable	
			OR (95% CI)	P value	
Respiratory failure	43	208			
Yes	18 (41.9)	47 (22.6)	2.52 (1.16–5.48)	0.020	
No	25 (58.1)	161 (77.4)	Reference		
ICU entrance	43	208			
Yes	11 (25.6)	24 (11.5)	2.61 (1.24–5.49)	0.012	
No	32 (74.4)	184 (88.5)	Reference		
Acute heart injury	43	208			
Yes	4 (9.3)	12 (5.8)	1.5 (0.44–5.1)	0.512	
No	39 (90.7)	196 (94.2)	Reference		
Acute kidney injury	43	208			
Yes	2 (4.7)	6 (2.9)	1.33 (0.25–7.02)	0.739	
No	41 (95.3)	202 (97.1)	Reference		
Septic shock	43	208			
Yes	4 (9.3)	6 (2.9)	3.31 (0.82–13.29)	0.091	
No	39 (90.7)	202 (97.1)	Reference		
DIC	93	565			
Yes	42 (45.2)	60 (10.6)	7.16 (4.33–11.85)	<0.001	
No	51 (54.8)	505 (89.4)	Reference		
Notes.

COVID-19 coronavirus disease 2019

PSM propensity score matching

ICU intensive care unit

DIC disseminated intravascular coagulation

The mortality rate was increased from 18.4% (134/729) in the non-thrombocytopenia group to 47.2% (60/127) in the thrombocytopenia group. After adjusting for age, sex, days from symptom onset to admission and comorbidities, the HR of the all-cause mortality in the thrombocytopenia group was significantly increased (HR 3.08, 95% CI [2.26–4.18], P < 0.001) compared with non-thrombocytopenia (Fig. 1B).

Comparison of haematological disorders based on the PSM cohort

The inter-group comparison was made at three clinical stages separately, i.e., 1–10 days, 11–20 days and 21–30 days after symptom onset, revealing that monocyte counts, monocyte percentage and basophil percentage were decreased in thrombocytopenia vs. non-thrombocytopenia patients at all three stages (Figs. 2E, 2F, 2I and Table S3). Moreover, significantly higher neutrophil counts and lower levels of lymphocyte percentage were observed for thrombocytopenia over non-thrombocytopenia group at 11–20 days and 21–30 days after symptom onset (Figs. 2B and 2C). Eosinophil counts and percentage at both 1–10 days and 21–30 days after disease onset were significantly lower in the thrombocytopenia group than in the non-thrombocytopenia group (Figs. 2H and 2G). The remaining indicators showed no inter-group difference, or only difference at the last clinical stage, which was less likely to be related to thrombocytopenia (Figs. 2A, 2D and 2J).

Figure 2 Dynamic profile of haematological indicators in groups of patients stratified by platelet counts on admission.

(A) Neutrophil percent; (B) neutrophil count; (C) lymphocyte percent; (D) lymphocyte count; (E) monocyte percent; (F) monocyte count; (G) eosinophil percent; (H) eosinophil count; (I) basophil percent; (J) basophil count. Dots present the values for individual detection, and the lines and error bars indicate the median and interquartile range, respectively. The dotted line indicates the lower limit and upper limit of each laboratory indicator. *P < 0.05 using a generalized estimating equation regression model adjusted for age, sex, delay from disease onset to hospital admission and comorbidities. The markers in comparisons without significant differences were not shown. The analysis was performed based on the PSM database.

Comparison of platelet morphology indexes and coagulation perturbation based on the PSM cohort

During the whole hospitalization, we evaluated platelet morphological indexes, which reflect the etiology of thrombocytopenia, and there were significant differences in two indexes between the two groups (Figs. 3A–3B). Mean platelet volume (MPV) increased, whereas the thrombocytocrit decreased, which corresponded with the reduction of platelet count.

Figure 3 Dynamic profile of platelet morphology indexes and coagulation factors in groups of patients stratified by platelet counts on admission.

(A) Mean platelet volume (MPV); (B) thrombocytocrit; (C) prothrombin time (PT); (D) activated partial thromboplastin time (APTT); (E) thrombin time (TT); (F) fibrinogen (FIB); (G) fibrinogen degradation products (FDP). Dots present the values for individual detection, and the lines and error bars indicate the median and interquartile range, respectively. The dotted line indicates the lower limit and upper limit of each laboratory indicator. P-values were calculated using generalized estimating equation regression models adjusted for age, sex, delay from disease onset to admission, and comorbidities. The markers in comparisons without significant differences were not shown. The analysis was performed based on the PSM database.

We also evaluated clotting profile and found a series of important coagulation factors with significant differences, including prothrombin time (PT), activated partial thromboplastin time (APTT), thrombin time (TT), fibrinogen (FIB), and fibrinogen degradation products (FDP) (Figs. 3C–3G). The thrombocytopenia group had higher levels of APTT and FDP at 1–10 days and 11–20 days after symptom onset, longer PT at 1–10 days, 11–20 days and 21–30 days, and higher level of TT at 11–20 days when compared with the non-thrombocytopenia group. Lower level of FIB was observed at 1–10 days and 11–20 days after symptom onset in the patients with thrombocytopenia.

Comparison of inflammatory markers and biochemical parameters based on the PSM cohort

Elevated levels of inflammation markers, including interleukin 6 (IL-6), tumor necrosis factor α (TNF-α), C-reactive protein (CRP) and procalcitonin, were observed at 1–10 days or 11–20 days after symptom onset in the patients with thrombocytopenia (Fig. 4). Especially for IL-6, the patients with thrombocytopenia had persistently elevated levels at 1–10 days, 11–20 days and 21–30 days after symptom onset, with their difference from the non-thrombocytopenia group further enlarged, which were significant at 1–10 days and 11–20 days after symptom onset. Higher level of CRP was demonstrated in patients with thrombocytopenia at 1–10 days, 11–20 days and 21–30 days after symptom onset, and with significant difference at 11–20 days compared to the non-thrombocytopenia group. The TNF-α and procalcitonin were elevated to higher levels in the thrombocytopenia group, with the difference attaining significance at 1–10 days compared to the non-thrombocytopenia group.

Figure 4 Dynamic profile of Inflammatory markers in groups of patients stratified by platelet counts on admission.

(A) Interleukin 6 (IL-6); (B) tumor necrosis factor α (TNF- α); (C) C reaction protein (CRP); (D) procalcitonin. Dots present the values for individual detection, and the lines and error bars indicate the median and interquartile range, respectively. The dotted line indicates the lower limit and upper limit of each laboratory indicator. P-values were calculated using generalized estimating equation regression models adjusted for age, sex, delay from disease onset to admission, and comorbidities. The markers in comparisons without significant differences were not shown. The analysis was performed based on the PSM database.

The levels of albumin (ALB) and total protein (TP) were reduced below normal range in the thrombocytopenia patients at 11–20 days or 21–30 days after symptom onset. The lactate dehydrogenase (LDH) was augmented above normal range for both groups, and to higher levels in the thrombocytopenia patients at 1–10 days and 11–20 days after symptom onset. The urea was exclusively elevated in the thrombocytopenia group, to a significantly higher level than in the non-thrombocytopenia group at 11–20 days and 21–30 days after symptom onset. Lower level of creatine kinase-MB (CK-MB) was observed at 21–30 days after symptom onset in the patients with thrombocytopenia (Fig. 5).

Figure 5 Dynamic profile of biochemical indicators in groups of patients stratified by platelet counts on admission.

(A) Total protein (TP); (B) albumin (ALB); (C) urea; (D) lactate dehydrogenase (LDH); (E) creatine kinase MB (CK-MB). Dots present the values for individual detection, and the lines and error bars indicate the median and interquartile range, respectively. The dotted line indicates the lower limit and upper limit of each laboratory indicator. P-values were calculated using generalized estimating equation regression models adjusted for age, sex, delay from disease onset to admission, and comorbidities. The markers in comparisons without significant differences were not shown. The analysis was performed based on the PSM database.

Demographic information, outcomes and laboratory findings of patients with late development of thrombocytopenia

Altogether 61 patients had late development of thrombocytopenia (the first observation of thrombocytopenia seen beyond 14 days post-symptom) (Table S4). Altogether 184 developed early thrombocytopenia (the first observation of thrombocytopenia seen within 14 days post-symptom). These two groups added up to a total frequency of 11.1% during the whole disease progress. Comparison between the two groups revealed older age with non-significant difference, significantly longer delay between disease and hospital admission, as well as longer duration of hospitalization in the late-thrombocytopenia group. The patients with late development of thrombocytopenia demonstrated higher levels of neutrophil counts at 21–30 days, CRP at 11–20 days and 21–30 days after symptom onset compared with patients with early development of thrombocytopenia (Figs. S2A and S2E). Lower levels of lymphocyte counts, INR and procalcitonin were demonstrated from the patients with late development of thrombocytopenia (Figs. S2B, S2D and S2F). Thrombocytocrit had a higher level in late development of thrombocytopenia group at 1–10 days and 11–20 days after symptom onset (Fig. S2C).

The nadir of PLT counts was observed at three weeks after disease for early thrombocytopenia and late thrombocytopenia (Fig. S3A). The all-cause mortality rate was comparable between early thrombocytopenia and late thrombocytopenia; however, survival analysis revealed a lower mortality rate at 21 days post symptom onset in the patients with late thrombocytopenia than early thrombocytopenia (21.3% vs. 33.7%), which trend, however, was reversed from 28 to 56 days post symptom onset (30.8% and 5.4% for patients with late and early thrombocytopenia, respectively) (Fig. S3B). This difference remained significant after adjusting the effect from age, sex, days from symptom onset to admission and comorbidities at 1–28 days (HR 0.43, 95% CI [0.24–0.75], P < 0.001) and 28–56 days (HR 4.89, 95% CI [1.91–12.55], P = 0.001).

Discussion

Platelets are the smallest enucleated blood cells at the center of coagulation and homeostasis, acting as the first participants in primary coagulation and playing an important role in the pathogenesis of thrombosis. Platelets are also part of innate immunity, which according to most recent studies, are widely involved in inflammation and immune responses and may form a bridge between the inflammatory and the coagulation system (Burzynski et al., 2019; Mancuso & Santagostino, 2017). Prior to this study, there was evidence suggesting thrombocytopenia to be a potential risk factor for the mortality of COVID-19, nevertheless, the thrombocytopenia can be used as a clinical biomarker for COVID-19 are still controversial (Amgalan & Othman, 2020). Moreover, the role of thrombocytopenia in COVID-19 pathogenesis and its relationship with systemic host response remained obscure. Lippi et al. published a meta-analysis study including nine studies with a total of 1779 COVID-19 patients. It revealed that platelet count was significantly lower in patients with more severe COVID-19 and was deemed associated with an increased risk of severe disease and mortality in patients with COVID-19 (Lippi, Plebani & Henry, 2020). In our study, based on a retrospective cohort study of COVID-19 patients, we comprehensively evaluated the haematological indicators, platelet morphology indexes, coagulation factors, inflammatory and biochemical markers in relation to thrombocytopenia at different stages of COVID-19. We suggest that thrombocytopenia might play an important role in the COVID-19 clinical outcome through a complicated process that was driven by its close relation to the haematological and coagulopathy progress, as well as inflammation response, which might be inducible to late developed DIC and respiratory failure.

Based on previous studies and our study, the proportion of thrombocytopenia as a clinical feature of COVID-19 was low, and bleeding is uncommon because the haemostatic balance is shifted markedly towards thrombosis (Tang et al., 2020). The need for transfusion in COVID-19 patients is not high and the diagnosis of thrombocytopenia is often overlooked in COVID-19.

There appears to be some variability in the percentage of COVID-19 patients who present with thrombocytopenia. For example, in a study analyzing data on 1,099 laboratory-confirmed COVID-19 patients from 552 hospitals in China, the patients with severe infection had 57.7% thrombocytopenia vs. 31.6% in less significant COVID-19 symptoms (Guan et al., 2020). Tang et al.’s (2020) study reported thrombocytopenia with a similar percentage of 57.1% in the non-survivors. In a series of COVID-19 patients from France, 24.9% of patients with thrombocytopenia were identified (Maquet et al., 2020). Chen et al. (2020) analyzed 99 COVID-19 patients, showing a much lower percentage of 12% with thrombocytopenia, and interestingly 4% with thrombocytosis. A meta-analysis of pooling data from totally 31 studies showed that thrombocytopenia had been reported in a higher proportion of severe patients vs. non-severe COVID-19 patients (Danwang et al., 2020). The prevalence of thrombocytopenia in the current case series was lower than most of previous studies. According to our study, the frequency of thrombocytopenia differed according to the clinical stages, when only 8.3% developed early thrombocytopenia, while 2.8% developed late thrombocytopenia, which might partially explain the variance among studies. In addition, our definition of thrombocytopenia is different from other studies. Other studies defined thrombocytopenia as PLT less than 125 or 150 × 109/L, while our definition of thrombocytopenia was less than 100 × 109/L. This more stringent definition may lead to fewer patients being included in thrombocytopenia group.

The reduced PLT counts act as an important indicator of adverse disease. At least three-fold increased risk of death and an increased risk of respiratory failure, ICU entrance and DIC were associated with lower PLT counts when the effects from age, sex and other important risk factors were accounted. There is an obvious dose effect of PLT on numerous important laboratory indicators, such as lymphocyte, LDH, ALB, IL-6, thrombocytocrit, indicating the important role of PLT participating in the anti-SARS-COV-2 host response. Moreover, the timing of thrombocytopenia development and fatal was correlated, as those with late thrombocytopenia tended to have late death than those with early thrombocytopenia.

To date, there are various mechanisms of viral-induced thrombocytopenia, which differ dependent on the virus types. A study conducted by Xu et al. proposed the following possible mechanisms: (1) direct infection of bone marrow cells by the virus and inhibition of platelet synthesis. Following virus infection, cytokine storm destroys bone marrow progenitor cells and leads to the decrease of platelet production. Lung injury indirectly results in reduction of platelet synthesis; (2) platelet destruction by the immune system; (3) platelet aggregation in the lungs, resulting in microthrombi and platelet consumption (Xu, Zhou & Xu, 2020). In addition, there might be potential platelet activation that was related to the thrombocytopenia, which is likely contributory to the hypercoagulability but the extent to which it does and which signaling pathways are involved warrant research. However, the exact mechanism of SARS-COV-2 related thrombocytopenia is far from clear. Further research is needed to clarify the mechanism

The current study was not designed to explore the pathophysiological mechanisms, however, we indeed revealed significant association that was driven by decreased PLT on the adverse COVID-19 outcome. In patients with thrombocytopenia, coagulation dysfunction involving PT, APTT, TT, FIB and FDP have been determined in the current study and reported in the previous studies (Guan et al., 2020). It’s also notable that more deviation from the normal value was observed for the coagulation factors, especially at 11–20 days after symptom onset. This time corresponded with the deterioration phase of the detrimental cases, in which DIC acted as a hallmark manifestation, which might suggest a significant role of severity of thrombocytopenia in the risk of extrinsic coagulation activation and DIC development. At the same time, a dysregulated immune response caused by decreased PLT is likely also be responsible for the increased disease severity of COVID-19, as a higher ratio of lymphopenia and increased levels of neutrophils, serum CRP, and IL-6 were observed in the patients with COVID-19 and thrombocytopenia observed at the early infection according to our study. The cytokines, such as IL-6 etc., may have higher levels in the serum than in the plasma, which may represent a higher expression state of systemic cytokine milieu. All these commonly seen abnormalities may account for the propensity of poor outcomes in patients with COVID-19 and early thrombocytopenia.

Disturbance of coagulation and fibrinolysis were also observed in acute COVID-19 infection (Paessler & Walker, 2013). Plasma PT and APTT are indexes of extrinsic and intrinsic coagulation system during the coagulation cascade, respectively (Chua et al., 1993). In the current research, we found a significant prolongation of both indicators occurring in the groups with thrombocytopenia. All these findings, taken together, suggested a perturbed extrinsic pathway in all COVID-19 patients and an even stronger activation of the intrinsic pathway in the severe cases (Mairuhu et al., 2003). Increased concentration of D-dimer indicated the activation of fibrinolysis, which is believed to be related to disseminated intravascular coagulation (Wada et al., 1993), a feature commonly seen in the severest stage of SARS-COV-2 infection. Platelet, although is the smallest of blood cells, have a central role in haemostasis regulation (George, 2000). The morphology and size parameters (MPV, thrombocytocrit and PDW) were the valuable reflections of platelet function (Thompson et al., 1982), both were also extensively comprised in the process of thrombocytopenia and contributed to the fatal outcome and severity of SARS-COV-2 infection.

These findings also offered alternative therapy choices of platelet transfusion for COVID-19 patients with thrombocytopenia, which was supported by the current finding that no detrimental effect was seen from the patients with slightly higher PLT counts above the normal range. Still, the usage of platelet transfusion for the SARS-COV-2 infection therapy should be cautiously applied. On the other hand, considering that thrombocytopenia involved not only reduction in PLT counts, but also platelet morphological changes, thus the mere platelet transfusion might not be efficient in reducing the adverse outcome.

The study was subject to major limitations that although dose–effect relationship of platelet counts and platelet hyperactivation, as well as the host immune response were observed, the causal relationship cannot be inferred. Whether thrombocytopenia is one of the causations of systematic host response, or only a representation of COVID-19 patients’ condition requires further clinical and experimental data to verify. Another limitation is that the drugs for the therapy of COVID-19 during the hospitalization may have a bias on the results in this study.

Conclusions

In conclusion, thrombocytopenia is associated with an increased risk of adverse outcomes in COVID-19 patients, and platelet count and related indicators should be included in the evaluation of COVID-19 patients during hospitalization together with other hematological, biochemical, and inflammatory indicators.

Supplemental Information

Supplemental Information 1 Supplementary Figures and Tables

Click here for additional data file.

Supplemental Information 2 Raw data

Click here for additional data file.

The authors would like to thank all the subjects, their families, and collaborating clinicians for their participation.

Additional Information and Declarations

Competing Interests

Author Contributions

Human Ethics

Ethics

Data Availability

The authors declare there are no competing interests.

Yang Yuan analyzed the data, prepared figures and/or tables, authored or reviewed drafts of the article, and approved the final draft.

Gang Wang analyzed the data, prepared figures and/or tables, authored or reviewed drafts of the article, and approved the final draft.

Xi Chen performed the experiments, authored or reviewed drafts of the article, and approved the final draft.

Xiao-Lei Ye analyzed the data, authored or reviewed drafts of the article, and approved the final draft.

Xiao-Kun Li analyzed the data, prepared figures and/or tables, authored or reviewed drafts of the article, and approved the final draft.

Rui Li performed the experiments, authored or reviewed drafts of the article, and approved the final draft.

Wan-Li Jiang performed the experiments, authored or reviewed drafts of the article, and approved the final draft.

Hao-Long Zeng performed the experiments, authored or reviewed drafts of the article, and approved the final draft.

Juan Du performed the experiments, analyzed the data, authored or reviewed drafts of the article, and approved the final draft.

Xiao-Ai Zhang analyzed the data, authored or reviewed drafts of the article, and approved the final draft.

Hao Li analyzed the data, authored or reviewed drafts of the article, and approved the final draft.

Li-Qun Fang analyzed the data, authored or reviewed drafts of the article, and approved the final draft.

Qing-Bin Lu conceived and designed the experiments, analyzed the data, prepared figures and/or tables, authored or reviewed drafts of the article, and approved the final draft.

Wei Liu conceived and designed the experiments, analyzed the data, prepared figures and/or tables, authored or reviewed drafts of the article, and approved the final draft.

The following information was supplied relating to ethical approvals (i.e., approving body and any reference numbers):

The study was approved by the Research Ethics Commission of Tongji Hospital, Tongji Medical College.

The following information was supplied relating to ethical approvals (i.e., approving body and any reference numbers):

the Research Ethics Commission of Tongji Hospital, Tongji Medical College.

The following information was supplied regarding data availability:

The raw data contains the variables used in the paper.

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
