# Peer review of "Thrombocytopenia and increased risk of adverse outcome in COVID-19 patients"

_PeerJ, doi:10.7717/peerj.13608_

## Round 0.1 · original submission · Major Revisions

Both reviewers advised major revisions, but if you revised correctly, it could be accepted.

Reviewer 1 ·

Basic reporting

This is an ambiguous report, however, the authors should revise the English style.
References are ok.
Article structure is correct.
The hypotheses proposed are also correct.

Experimental design

This is a retrospective observational study in which platelet counts were the guide biomarker.

Validity of the findings

Other studies have reported on the importance of thrombocytopenia as a marker of worse outcomes in patients with covid.

Additional comments

This is a multi-centre retrospective study with the aim of evaluating the relationship between thrombocytopenia (defined as platelet count < 100 x 109/L) and disease progression. The outcomes were acute respiratory distress syndrome (ARDS), respiratory failure, intensive care unit (ICU) admission, acute heart injury, septic shock, and disseminated intravascular coagulation (DIC).
The authors included 2,209 patients, of which 127 (5.7%) had thrombocytopenia. Thrombocytopenic patients were older, however, the authors created a matched non-thrombocytopenic group according to age (Table 1).

Criticisms
• The English style is clear and unambiguous, however, the authors should choose American or British style of spelling, for instance, haematological or hematological, as both are represented in the manuscript.

Results section:
• lines 176-178, “While thrombocytopenia is a key diagnostic component in DIC, its association with DIC is understandable”, are not clear. Please, rephrase them. The same could be asserted regarding the subsequent lines (178-180), in which the authors state that there was no difference between the two groups regarding some variables, in contrast to what was stated at the beginning of this subsection.
• Figure 2- I believe it is unnecessary to show percent of cell types, as it overloads the figure (and the message).
• I think that there are too many figures. The authors could have selected only the altered variables to show in the figures.
Conclusions section can be shortened.

Reviewer 2 ·

Basic reporting

Clear and unambiguous, professional English used throughout.
Literature references, sufficient field background/context provided.
Professional article structure, figures, tables. Raw data shared.
Self-contained with relevant results to hypotheses.

Experimental design

Original primary research within Aims and Scope of the journal.
Research question well defined, relevant & meaningful.
Rigorous investigation performed to a high technical & ethical standard.
Methods described with sufficient detail & information to replicate.

Validity of the findings

Impact and novelty not assessed. Meaningful replication encouraged where rationale & benefit to literature is clearly stated.
All underlying data have been provided; they are robust, statistically sound, & controlled.
Conclusions are well stated, linked to original research question & limited to supporting results.

Additional comments

Dear Editor,

PeerJ.

Subject: Peer review

I had the pleasure of reviewing the manuscript submitted to your journal, entitled ‘The association between thrombocytopenia and systemic disorders in COVID-19 patients’

The paper needs further major revisions. The paper would be strengthened by:

Major points:
1. The title ‘systemic disorders’ is misinforming as the paper focuses on clinical outcomes and laboratory values differences among COVID-19 patients with and without thrombocytopenia.
2. The case to control ratio in propensity match is 1:6. 127 patients in thrombocytopenia and 729 in the non-thrombocytopenia group do not precisely add up to the ratio (127*6=762).
3. Expand ‘ISTH’ for DIC score.
4. The term acute heart injury (Table 1) is non-specific. Please clarify.
5. Figures 2, 3, 4, 5 need P-values as the visual discernment between two groups may be difficult. The P values may be added within the figure for each set of measurements.
6. Lines 211-215 are most suitable under discussion rather than the results section.
7. Lines 253-261 talk about ‘both groups.’ Please mention that the two groups are early thrombocytopenia vs. late thrombocytopenia for better understanding.
8. Explore palusble reasons why the low rates of thrombocytopenia were observed in this cohort compared to other studies on this topic.
9. Cite the paper by Panyang Xu about the pathogenesis of thrombocytopenia in COVID-19. Ann Hematol. 2020; 99(6): 1205–1208. Published online 2020 Apr 15. doi: 10.1007/s00277-020-04019-0 PMCID: PMC7156897 PMID: 32296910.

Minor points:
1. Line 171, the phrase ‘higher prevalence of ARDS’ is more appropriate than ‘frequent occurrences of ARDS’.
2. The phrase PSM database is misleading. Recommend using 'PSM cohort' for better understanding.
3. Are ARDS and respiratory failure in Table 2 mutually exclusive?

---

## Round 0.2 · accepted · Accept

Well revised following the advice.

Reviewer 1 ·

Basic reporting

This is a multi-centre retrospective study with the aim of evaluating the relationship between thrombocytopenia (defined as platelet count < 100 x 109/L) and disease progression. The outcomes were acute respiratory distress syndrome (ARDS), respiratory failure, intensive care unit (ICU) admission, acute heart injury, septic shock, and disseminated intravascular coagulation (DIC).
The authors included 2,209 patients, of which 127 (5.7%) had thrombocytopenia. Thrombocytopenic patients were older, however, the authors created a matched non-thrombocytopenic group according to age

Experimental design

This is a multicentre retrospective study in which more than 2,000 hospitalised patients were included.

Validity of the findings

Yes, the findings are well grounded. The results are not exactly new, however, they are sound and useful to the physicians and researchers on this area. Patients with thrombocytopenia had worse outcome.

Additional comments

No comments.